# ‘Brave Enough’: A Qualitative Study of Veterinary Decisions to Withhold or Delay Antimicrobial Treatment in Pets

**DOI:** 10.3390/antibiotics12030540

**Published:** 2023-03-08

**Authors:** Ri O. Scarborough, Anna E. Sri, Glenn F. Browning, Laura Y. Hardefeldt, Kirsten E. Bailey

**Affiliations:** 1Asia-Pacific Centre for Animal Health, Melbourne Veterinary School, University of Melbourne, Parkville, VIC 3010, Australia; 2National Centre for Antimicrobial Stewardship, Peter Doherty Institute, Parkville, VIC 3052, Australia

**Keywords:** antibiotic, resistance, stewardship, animal, One Health, barriers, enablers, prescribing, behaviour, TPB

## Abstract

Veterinarians sometimes prescribe antimicrobials even when they know or suspect that they are unnecessary. The drivers of this behaviour must be understood to design effective antimicrobial stewardship interventions. Semi-structured interviews were conducted with 22 veterinarians who treated companion animals in Australia. The Theory of Planned Behaviour was used to organise interview themes, focusing on a decision to withhold antimicrobial therapy in the absence of a clear indication. Many background factors influenced antimicrobial-withholding decisions, including the veterinarian’s communication skills, general attitudes towards antimicrobial resistance (AMR), habits and energy levels. Client awareness of AMR and the veterinarian–client relationship were also important. Beliefs about the consequences of withholding antimicrobials (behavioural beliefs) were dominated by fears of the animal’s condition deteriorating and of failing to meet client expectations. These fears, weighed against the seemingly distant consequences of AMR, were major barriers to withholding antimicrobials. Normative beliefs were primarily focused on the expected approval (or disapproval) of the client and of other veterinarians. Control beliefs about the difficulty of withholding antimicrobials centred around client factors, most importantly, their capacity to adequately monitor their animal, to pay for further investigations, or to undertake non-antimicrobial management, such as wound care, at home. The use of antimicrobials by companion animal veterinarians in the absence of a clear indication is often powerfully driven by behavioural beliefs, chiefly, fears of clinical deterioration and of failing to meet client expectations.

## 1. Introduction

Antimicrobial resistance (AMR) is a major threat to environmental, animal and human health. In bacteria, AMR occurs as a result of the survival and proliferation of resistant bacterial subpopulations, and the transfer of AMR genes between bacteria [1]. This process is accelerated under the selection pressure of antimicrobial agents [2]. AMR bacteria can lead to serious adverse outcomes for the treated individual [3], in-contact animals (including humans) [4], and wider animal populations [5,6]. To reduce morbidity and mortality attributable to AMR and to preserve the effectiveness of existing antimicrobials for as long as possible, prescribers should use antimicrobials only when they are needed and when they are likely to improve the health outcome of the patient. However, there is a significant body of evidence showing that prescribers in all sectors are still using antimicrobials in clinical syndromes where they are unlikely to be helpful, selecting empirical antimicrobials that have a broader spectrum of activity than is recommended in treatment guidelines and prescribing antimicrobials for longer durations than necessary [7,8,9,10].

Although such prescribing can be driven by a limited understanding of microbiology and pharmacology, studies suggest that emotional reasons, socio-cultural factors and practical constraints often play a greater role in antimicrobial prescribing decisions than a lack of prescriber knowledge [11,12,13]. In other words, prescribers sometimes do things they are aware are not best practice. Despite the importance of qualitative research in understanding the drivers of suboptimal antimicrobial use [14], there are only four qualitative studies and one mixed-methods study of antimicrobial prescribing decisions by companion animal veterinarians in the English-language scientific literature [15,16,17,18,19], all conducted in the United Kingdom and the Netherlands. Australian veterinarians could have somewhat different approaches to antimicrobial use than their British and Dutch counterparts, due to differences in veterinary training, antimicrobial susceptibility of common pathogens, the availability and regulatory status of particular drugs, and the socio-economic, geographical and cultural context, including a much lower rate of pet health insurance compared with the UK [20,21]. A previous qualitative study on barriers to antimicrobial stewardship in Australian veterinary practices [22] revealed some of the non-clinical reasons Australian veterinarians used antimicrobials, providing a foundation for this in-depth interview study.

The framework of Ajzen’s Theory of Planned Behaviour (TPB) [23] was used to structure the thematic analysis because it was felt to most accurately reflect the relationship between the contributory factors expressed by participants. The TPB has been previously applied in studies of prescribing behaviour [24,25,26,27,28,29]. The TPB postulates that three groups of beliefs contribute to a person’s intention to perform a behaviour, and that the strength of this intention is correlated with the likelihood of performing it. These groups of beliefs are: (1) behavioural beliefs, i.e., beliefs about the consequences of a behaviour and moral and ethical considerations of these consequences; (2) normative beliefs, i.e., perceived social pressure; and (3) control beliefs, i.e., the perceived barriers to and enablers of a behaviour, which lead to the perceived ease of executing it [23] (Figure 1). It is important to note that a person’s beliefs need not be formed through rational or even conscious thought [30] and that they are influenced by background factors, such as gender and knowledge.

The primary objective of this phenomenological, qualitative study was to explore the drivers of potentially unnecessary use of antimicrobials in Australian companion animal practice, by examining the decision to withhold or delay antimicrobial use. A secondary objective was to explore veterinarians’ suggestions about how they could be better supported to withhold or delay use of antimicrobials, where clinically appropriate.

## 2. Materials and Methods

In-depth, semi-structured one-on-one interviews were conducted with veterinarians registered in Australia who treat small companion animals. Most participants were recruited through two closed social media groups for Australian veterinarians, with purposive sampling of additional veterinarians approached through the authors’ personal networks, to ensure inclusion of veterinarians across a broad demographic range and with diverse levels of interest in antimicrobial stewardship (AMS). None of the participants worked in the same practice as any of the other participants. Interview participants were offered an AUD 50 supermarket voucher as a token of appreciation for their time.

Two closed social media groups for Australian-registered veterinarians were used to recruit the majority of interviewees. Additional veterinarians were approached through the authors’ personal networks to ensure inclusion of veterinarians across a broad demographic range and with diverse levels of interest in AMS.

### 2.1. Pre-Interview Survey

Participants completed a short online survey in Qualtrics (www.qualtrics.com, accessed on 20 March 2020) to gather demographic and professional role information and understand the veterinarian’s level of interest in antimicrobial resistance and stewardship. The pre-interview survey also asked participants to estimate the relative contributions of five different sectors (human hospitals, human primary practice, companion animals, livestock and other) to antimicrobial resistance in Australia. This question was used to gauge the participants’ views of their own sector’s contribution to AMR. Informed consent was obtained from all subjects involved in the study via the survey.

### 2.2. Interviews

In-depth, semi-structured one-on-one interviews were conducted by R.O.S., a veterinarian, in June and July 2020, using a video teleconferencing platform (www.zoom.us, accessed on 5 June 2020). The early part of each interview included the discussion of two clinical vignettes for which national antimicrobial use guidelines exist: (a) a 2-year-old German Shepherd dog presenting with signs of lower urinary tract infection; and (b) an 8-year-old domestic short hair cat presenting with a cat fight abscess. The interview guide is provided in Appendix B. Participants were asked to explain the investigations and treatment they wished to undertake and why, and how they would respond to various additional pieces of information (e.g., what if the cat’s owner says she cannot give tablets?) or alterations to the case story presented (e.g., what if the dog had already presented twice in the last three months with the same clinical signs?). The remainder of the interview was highly adaptive and guided by the participant’s earlier responses. Interview participants were offered an AUD 50 supermarket voucher as a token of appreciation for their time.

Participants were interviewed until three consecutive interviews had not generated important new themes. Interviews were recorded using the video teleconferencing software, transcribed using online transcription software (otter.ai, accessed on 5 June 2020), and manually corrected by one author (R.O.S.). All identifying data, such as names of veterinary practices, colleagues and towns were redacted. An inductive coding approach was used. Two authors (R.O.S., A.E.S.) independently coded the same interview according to the draft codebook, using NVivo12 (QSR International, 2020). Discrepancies in coding were discussed and the codebook refined. A second interview was then coded by both researchers in tandem to finalise the interview themes. Coding of all remaining interviews was completed by R.O.S. Themes related to a decision to withhold or prescribe antimicrobials were subsequently reorganised within the Theory of Planned Behaviour (Figure 1), focusing on the behaviour of withholding (or delaying) antimicrobial treatment. The final codebook is in Appendix C.

## 3. Results and Discussion

The median interview duration was 90 min (IQR 73-98). Four of the 22 participants were previously known to the interviewer.

### 3.1. Pre-Interview Survey (Participant Characteristics)

The 22 interviewed veterinarians represented a broad range of relevant demographics. Participant characteristics are shown in Table 1.

Their median estimate of the companion animal sector (including horses) contribution to the overall problem of AMR in Australia was 15% [range 0–25%] (Table A1, Appendix A).

### 3.2. Interviews

The median interview duration was 90 min (IQR 73-98). Four of the 22 participants were previously known to the interviewer.

Where a theme was raised by more than one participant, non-specific semi-quantitative words such as ‘some’ or ‘multiple’ are used. Enumeration is reserved for responses to the case studies, because these were discussed consistently with every participant. However, no inferences should be drawn from a qualitative study about the prevalence of a phenomenon beyond the sample group.

An overview of interview findings is presented in Figure 2.

#### 3.2.1. Background Factors

Participants identified several important background factors (Figure 3) that influenced a decision to withhold or delay antimicrobial treatment in the absence of a clear clinical indication.

Several participants expressed a high degree of concern about antimicrobial resistance, as could be expected of a mostly self-selected cohort. They expressed fear of the consequences of AMR (a1) and a sense that the veterinary profession broadly (a2, a4) and that they personally (a2, a3, a4) bear some responsibility for perpetuating AMR. These attitudes were also reflected in a previous survey of Australian veterinarians, doctors and dentists [31]. Some participants in our study also reflected on the contribution of antimicrobial use in humans to the broader problem and felt that veterinarians were unfairly blamed for AMR problems in people (a5, a6).

AMR can seem like a distant threat, even though every use of an antimicrobial selects for resistant bacteria, which can persist in the animal for months to years [32] and present a risk to veterinary staff, clients and future patients [33]. Multiple veterinarians described the trade-off they made between the invisible risk of AMR and their more immediate concerns of sick pets and angry clients (a8, a9). This tension was also described by UK companion animal veterinarians [15,17]. Tellingly, one participant who worried that AMR would lead to animal infections she would no longer be able to treat (a2) also indicated that, in a case where it was uncertain whether antimicrobials would be helpful, she was willing to accept the risk of selecting for AMR—and other risks associated with antimicrobial use—for even a minor increase in the probability of clinical resolution, because it was more important to her to ‘fix’ an animal as quickly as possible (a10). This pressure to resolve a problem in a single visit was also described by other participants (b14, c1, n1, n18) as a barrier to withholding antimicrobial treatment. It was also evident in studies of veterinarians in the UK [15] and The Netherlands [19], and is supported locally by a survey study of Australian pet owners, one-quarter of whom indicated that they would be annoyed if their animal was not fixed the first time and they had to return to the clinic [34]. However, a search of the medical literature failed to find a similar expectation in human medicine. Further research is warranted to explore the reasons behind this attitude among pet owners, whether it occurs in human patients and whether it can be mitigated by improved communication by prescribers.

Interestingly, only one of the participants (a1) mentioned the direct adverse effects that systemic antimicrobials commonly have on patients, such as nausea, diarrhoea and lingering disruption of the animal’s microbiome, as additional reasons to avoid unnecessary antimicrobial use. One veterinarian even stated that there was no direct, immediate harm from antimicrobials (c6). Reminding veterinarians—and pet owners—of these additional risks may increase their motivation to avoid antimicrobial therapy.

One participant with a low interest in AMR and AMS demonstrated circular reasoning that might explain complacency about AMR among some veterinarians. Because she was not seeing many AMR organisms in her practice—organisms that can only be identified by bacterial culture and susceptibility testing—she felt that AMR was not yet a serious problem in animals. Yet she also acknowledged that she and her colleagues requested very few culture and susceptibility tests (a7). This suggests that some veterinarians will only recognise the risk of AMR when they observe antimicrobial treatment failure.

Some veterinarians mentioned common clinical scenarios that were previously thought to require antimicrobials, and for which there is now well-publicised evidence that antimicrobials are not indicated, including acute diarrhoea in dogs [35], dental surgeries in systemically well patients and routine desexing surgeries [36,37]. Some noted that they themselves—or other veterinarians they knew—persisted in giving antimicrobials in such cases, and that habit (a11) was a contributor to that choice. UK veterinarians also acknowledged the role of habit in their antimicrobial prescribing [16].

High workloads and a related lack of energy (a12) were also recognised as contributing to decisions to prescribe when there was no clear indication, especially in combination with a client who appeared to expect antimicrobials. Increasing veterinarian workloads, partially due to increasing rates of pet ownership during the COVID-19 pandemic, and attrition from the veterinary clinical workforce [38,39] may therefore increase inappropriate antimicrobial prescribing. The effect of fatigue was echoed in a Danish study of medical general practitioners, who were 25% more likely to prescribe antimicrobials in their fourth than in their first hour of consulting [40]. Being time-poor and lacking access to paywalled scientific journals also inhibited veterinarians from staying up to date with current antimicrobial use recommendations (a13). Considering this, researchers hoping to influence veterinary antimicrobial prescribing are likely to have more impact with open-access publishing, and by providing easy-to-digest study summaries for clinicians.

Many of the more experienced veterinarians reflected on the change in their propensity to prescribe antimicrobials between the beginning of their career and the current time, due to the development of confidence and communication skills. Clinical experience and additional training could give veterinarians assurance that certain common illnesses will resolve without antimicrobial treatment (a14). The development of assertiveness and strong communication skills—to explain to an expectant client why antimicrobials are not required—also enabled a decision to withhold or delay prescription of antimicrobial treatment for an animal (a14, a15, a16, a17). These findings echo those of a UK study, which also suggested that communication training for veterinarians could enable AMS [17]. However, regardless of confidence and communication skills, the prejudices some clients held about younger (a18, a19) and female (a19) veterinarians made it more difficult to convince them that antimicrobials were not needed.

One veterinarian and practice owner noted that a workplace culture that is receptive to change had been a powerful enabler of avoiding unnecessary antimicrobials. In this practice, actively seeking and fairly evaluating the input of all staff had led to the elimination of routine antimicrobial use for desexing and other clean surgeries (a20), in line with current veterinary antimicrobial use guidelines [41]. Mateus et al. in the UK also noted that regular meetings where veterinary staff discuss clinical cases and protocols was an enabler of AMS [15]. Similarly, a supportive workplace culture—where veterinarians had confidence their colleagues would support their decisions (a12)—was noted to enable withholding of antimicrobial treatment.

Several important background factors relating to a practice’s clientele were also identified. High socio-economic advantage (a22), pet insurance uptake (a23) and high health literacy (a24, a25) were all enablers of decisions to withhold or delay antimicrobial treatment. These were felt to increase the client’s willingness and capacity to pursue (often expensive) diagnostic tests rather than (cheaper) antimicrobial treatment trials, or to undertake conservative management and monitoring of the animal at home and return to the clinic if needed. One veterinarian in a regional town also acknowledged the positive influence of the local medical general practitioners, who over the years had educated their patients about the importance of avoiding unnecessary antimicrobial treatment. The understanding this had established in the local community had made it easier for the veterinarians to withhold or delay antimicrobial treatment for the same people’s pets (a26). That people are applying the information they receive from their own doctors to their pets’ health care suggests that the reverse might also be true, and supports collaborative medical–veterinary efforts to provide consistent messages to the public about responsible antimicrobial use, such as the ‘Antibiotic Guardian’ campaign that started in the UK [42].

The relationship between the veterinarian and the client was another key background factor in a decision to withhold antimicrobial treatment. When dealing with a ‘difficult’ client, it was tempting to avoid confrontation and dispatch them by prescribing a medication that the client perceived to be curative (a29), such as antimicrobials. In contrast, a ‘good’ client could work with the veterinarian to implement a non-antimicrobial management plan (a30), had realistic expectations of outcomes and was unlikely to become angry if the animal’s condition did not immediately improve (a31). Similarly, when the client expressed frustration or exhaustion (a27, a28) from managing the unwell animal at home (e.g., severe diarrhoea, coughing), the veterinarian felt more compelled to take tangible action, and that action was often to give antimicrobials.

Where withholding antimicrobials was supported by evidence-based antimicrobial prescribing guidelines (a32) and/or expert opinion, particularly that of registered veterinary specialists (a33), veterinarians felt more comfortable withholding antimicrobials. Quoting an external source of ‘truth’ could deflect client pressure to prescribe and could be particularly helpful for less experienced veterinarians (a32). This phenomenon of deferring to external sources was also described in a study of Dutch veterinarians [19]. Guidelines that recommend withholding antimicrobials for a particular condition would also protect the veterinarian if the client were later to lodge a complaint with the veterinary registration board. However, awareness of guidelines was patchy. In Australia, independently-developed veterinary prescribing guidelines had existed for companion animals for a few years prior to the interviews [41,43] but few participants had seen them, despite the presumed bias in this study towards veterinarians interested in AMS. This suggests that more effort is required to distribute and publicise the availability of guidelines for antimicrobial use within the Australian veterinary profession.

#### 3.2.2. Behavioural Beliefs

Behavioural beliefs, i.e., beliefs about the consequences of a behaviour and the implications thereof (Figure 4), featured strongly in a veterinarian’s decision to withhold antimicrobials when there was no clear indication. These beliefs were dominated by fear of the animal deteriorating clinically, and the related fear of failing to meet owner expectations.

Multiple interviewees described implicit and/or explicit client expectations of antimicrobials, similar to a qualitative study of companion animal veterinarians in the UK [17]. These experiences are also reflected in a recent study, in which 15% of Australian pet owners indicated that they had explicitly requested antimicrobials from their veterinarian [34]. However, a few interviewees in our current study felt that this was becoming less common and attributed this to increasing public understanding of AMR (n5, n6, a25). UK veterinarians also reported increasing public awareness of AMR as an enabler of withholding antimicrobials [16,17]. However, studies in both countries have found that public understanding of AMR in pets was still poor overall and knowledge of interspecies transmission of bacteria was low [17,34].

Some veterinarians felt that if they were to withhold antimicrobials, particularly with clients who expected antimicrobials and were known or suspected to be ‘difficult’ (a29) and demanding (a31), it would lead to a long, possibly adversarial (b20, b21, b22) conversation with the client, requiring time and energy that they rarely had. In such scenarios, many veterinarians felt that they had only two choices: to go bravely into ‘battle’ with that client (b21) and accept the possible fallout or capitulate. Given that this fallout could include clients verbally abusing them or their colleagues (b1, b15, b17), lodging a complaint about them with the veterinary board (b16), pursuing civil legal action (n12–17) or ‘slaughtering’ the clinic on social media (b17, b18), with impact on staff mental health and business reputation and revenue, it is clear why veterinarians sometimes felt that withholding antimicrobial treatment was simply ‘not worth the crusade’ (b15) and occasionally used antimicrobial therapy as an ‘easy way out’ (b20). There was also a sense that there might be no benefit in withholding antimicrobials, as a determined client could obtain antimicrobials from another veterinarian at the same practice (b24) or take their business to a different practice (a3, b26). Losing dissatisfied clients to another practice was a concern for some veterinarians and it was a bigger risk where there were many other veterinary practices in the local area (b26). However, some veterinarians were willing to stand by their principles, even if it meant losing clients (a3).

The second scenario was a known bacterial infection that could be managed without systemic antimicrobials, but where non-antimicrobial management would be more difficult for the client. There can be a higher risk of treatment failure if antimicrobials are withheld, due to the client failing to execute adequate home management, unrecognized lowered immunity in the animal, or the development of unforeseen infective complications. However, in some cases this risk may be managed by improving client instruction and follow-up, rather than by using systemic antimicrobials. For example, client handouts explaining non-antimicrobial management and automated text messages to check on progress.

One example of this is the cat fight abscess case study discussed with all participants; the draining subcutaneous abscess (without cellulitis) described is not life-threatening and would usually resolve without antimicrobials, provided that the cat has normal immune function and drainage is maintained by the client regularly cleaning the abscess, as recommended by Vet H (b2) and local prescribing guidelines [41]. However, most participants in this study (20/22) opted to give this cat systemic antimicrobials, most commonly amoxicillin–clavulanate for 5 to 7 days (12/22), or cefovecin (5/22), a single long-acting injection with a duration of efficacy of 14 days. These veterinarians were asked whether they would consider getting the owner to manage the wounds instead. Some said that there were rare, highly competent, clients who could be trusted with this task (c11, c12), but for all other clients, they would give antimicrobials to decrease the risk of treatment failure and thereby protect themselves from client dissatisfaction (b4). Some veterinarians used phrases that framed antimicrobials as a form of self-defence, such as ‘cover my bum’ (b4) or ‘cover my bases’ (b6). Previous adverse experiences with withholding antimicrobial treatment reinforced this behaviour (b5). Another admitted that, for her, giving antimicrobials was an irrational, almost involuntary, response to the sensory experience of pus (b3).

The third scenario is when the diagnosis was unclear and antimicrobials were given just in case they helped, as a ‘treatment trial’. This was particularly true where there were clinical signs that are associated with a bacterial infection, such as fever, but which could equally be caused by viral infection or even non-infectious conditions. This type of prescribing has also been described in medical intensive care units [44].

Antimicrobial treatment trials are relatively common in veterinary medicine for a few reasons. The non-verbal nature of the patients—and clients who are not always able to observe their animals—mean that the history and physical examination of an ill cat or dog are often insufficient to reach a diagnosis. At this point, there are three main pathways the veterinarian can take: diagnostic work-up; watch-and-wait; or a treatment trial—commonly, a course of antimicrobials. The latter approach can be appealing when the veterinarian weighs the invisible and seemingly distant consequences of AMR (a8, a9) against the drawbacks of the first two approaches.

Diagnostic work-up is often expensive for the client, except in the unusual instances where the animal is insured, and a definitive diagnosis is sometimes never achieved, despite costly investigations. There are also workflow implications for the clinic. Unlike human patients, who are usually sent away to specialist providers for investigations, investigations on veterinary patients usually occur within one general practice clinic. This can involve additional time and labour—for example, a fractious cat may need to be admitted to hospital, sedated and then handled by multiple staff, simply to obtain a blood or urine sample. Some investigations require specialised equipment, supplies or personnel that must be brought in from elsewhere. Hence, work-up can pose a challenge, particularly in a busy clinic, and a course of antimicrobials can be a way to avoid—or at least postpone—that additional work (b23). Similarly, UK and Dutch veterinarians felt that time pressure and cost to the client were barriers to performing appropriate investigations and enablers of antimicrobial treatment [15,19].

A watch-and-wait approach—which can include a ‘delayed’ antimicrobial prescription—is inexpensive and is commonly the most prudent course of action for an animal with mild, non-specific signs. Some participants said they often used this approach to keep an owner happy despite withholding antimicrobials (b19), but it poses a higher risk of clinical deterioration if a serious bacterial infection is indeed developing (b6, n17). Additionally, this approach sometimes requires the veterinarian—or other clinic staff—to expend time and energy coaching the client to provide supportive care and monitor changes in the animal’s condition. Some veterinarians said that they would feel compelled to call the client days later, in case antimicrobials (or some other treatment) needed to be initiated. When exhausted and under time pressure, this extra work associated with the watch-and-wait option becomes less appealing. Furthermore, the veterinarian must have faith that the client will not perceive watch-and-wait as the veterinarian ‘doing nothing’ (n10, b7) and hence not providing value for the consultation fee (b7, b11). They must also trust that the client has the capacity (a24) to carry out their recommendations, including contacting the clinic in a timely fashion if the animal needs further attention, and will not become irritated about returning to the clinic to collect medication. This trust is not always present (c11, c12).

Where the diagnosis was unclear and an animal had severe clinical signs, veterinarians were also less likely to withhold antimicrobials because the risk/benefit equation had shifted (b8, b9). This kind of ‘Hail Mary’ prescribing—hoping antimicrobials might be life-saving—was particularly appealing when the client was unable or unwilling to spend money on further diagnostic tests (b8).

Two major fears predominated when veterinarians considered withholding antimicrobials: the fear of clinical deterioration (b1, b2, b3, b4, b8, b9, c11) and the fear of failing to meet client expectations (b7, b12, b13, b15, b16, b17, b18, b26), and the downstream consequences of these two outcomes. These two fears—and the factors that heightened those fears—are shown in Figure 5, to further describe the logic and the connections between these central ideas described by interviewees.

The perceived impact of withholding antimicrobials was another important behavioural belief for veterinarians. While some interviewees saw every course of antimicrobials that they avoided as a victory, others saw it as futile, because they believed most other veterinarians were using antimicrobials liberally (b25) and considered their personal contribution to antimicrobial use to be negligible, or because they expected that the client would simply obtain the antimicrobials from another veterinarian (b24, b26). In a study of Australian medical general practitioners, there was also a sense of futility about withholding antimicrobials, but those doctors’ reasons were quite different; they saw human hospital and veterinary use of antimicrobials as dwarfing their own sector’s contribution [45].

#### 3.2.3. Normative Beliefs

Interviewees mentioned a range of people whose expectations influenced their decision to withhold antimicrobial treatment (Figure 6). Client expectations were commonly mentioned, but the expectations and modelled behaviours of their employer and colleagues, and the perceived expectations of the veterinary board, were also important. Somewhat surprisingly, the influence of veterinary academics was also mentioned by a few participants.

However, some veterinarians reported that expectations of antimicrobial treatment were uncommon in their clinic clientele (a25, n5). This could be due to historically conservative antimicrobial use by veterinarians at that clinic, the client’s level of understanding of AMR (n5), which can in turn be influenced by educational level (a24, a25) or, as mentioned above, by the AMS efforts of local medical general practitioners (a26). One veterinarian described the key role of reception or nursing staff in setting expectations of antimicrobials prior to the consultation (n4).

Interviewees were less likely to withhold antimicrobials if they suspected their employer and colleagues would judge them negatively for failing to fix the ailment the first time (n7), for not meeting client expectations (n8) or for falling short of revenue targets (n1). Some participants reflected on the lingering influence of an employer they had had early in their career, who had encouraged unnecessary antimicrobial use (n9, n10). Conversely, having veterinary colleagues who valued responsible antimicrobial use (a21) enabled withholding of antimicrobial treatment. Veterinary nurses who understood antimicrobial resistance and were ‘converted’ to the cause (n15) were also cited as powerful enablers, highlighting the valuable role of para-veterinary staff in AMS programs. Interviewees also mentioned that when veterinary specialists, who were viewed with reverence, condoned the withholding of antimicrobial treatment in particular conditions (n13), this helped them to do the same.

Older veterinarians and ‘old-school’ ways of practising veterinary medicine (n1, n8, a21) were mentioned by some younger interviewees in the context of more liberal antimicrobial use, and younger veterinarians were sometimes associated with more conservative use (a20, a21). This was also reported in two UK studies [15,16], but is somewhat contradicted by our finding that veterinarians with more clinical experience and stronger communication skills felt more confident in withholding unnecessary antimicrobials (a14, a15, a16). Indeed, one of the older veterinarians in our study believed that younger veterinarians prescribed more antimicrobials out of fear or a lack of confidence (b7). Notably, while the median age of interviewees was 34 years, the only two participants who indicated that they would not prescribe antimicrobials for the cat fight abscess case were aged 68 (b2) and 43. Quantitative research is needed to establish whether there is a true association between the clinical experience of veterinarians and unnecessary antimicrobial prescribing, and if there is, in which direction it influences prescribing.

Belief that there is widespread liberal (b6) or ‘willy nilly’ (a4) use of antimicrobials by the Australian veterinary profession was a barrier to withholding antimicrobials. Veterinarians cited a few reasons for this: firstly, the perceived likelihood that their withholding of antimicrobials would be futile, because it would have negligible impact on the overall problem (b25), and because the client might just obtain antimicrobials from another veterinarian (b24, b26). Veterinarians also feared the veterinary board (n16, n17) would rule against them if the case deteriorated, because professional negligence is judged by comparison with what ‘a reasonable colleague’ would do. If a veterinarian believes that most of their profession prescribes antimicrobials to dogs and cats at high rates, it follows that withholding antimicrobials is more likely to be viewed by a veterinary board as negligent (b9, b12, n14). Comparison of two recent AMS trials in the UK [48] and Australia [49] suggests that overall companion animal antimicrobial prescribing rates in Australia are probably relatively low. However, a central tenet of the TPB—and many other behavioural theories—is that it is a person’s belief about relevant others, rather than the evidence, that shapes behaviour. Indeed, prescribing antimicrobials to avoid the significant stress of a veterinary board hearing in itself—with the potential for professional reprimand—was mentioned by multiple interviewees (n16, b16, a29).

A few veterinarians felt that the academics who had taught them as undergraduates would disapprove of some of their current antimicrobial use. However, the other element of the subjective norm—how much these veterinarians cared about the approval of this group—was low. One indicated that the approval of the client who expected antimicrobials outweighed the academics’ disapproval ‘in the back of [her] head’ (n18). Similarly, another defiantly acknowledged that she was far more strongly motivated by protecting herself and her colleagues from client dissatisfaction than by the approval of veterinary academics (n19).

#### 3.2.4. Control Beliefs

Although a decision to withhold (or prescribe) antimicrobials is always within the control of the veterinarian, there were several situations in which veterinarians perceived that there was no practical alternative to prescribing antimicrobials (Figure 7).

The time that the animal is presented to the clinic can determine whether the veterinarian feels it is feasible to withhold antimicrobials. Both the time pressure in a busy clinic (b20), especially being ‘slammed’ on Fridays (c1) and Saturdays (b23, c2), and the lack of other clinic staff to assist with investigations (c2), tended to lower a veterinarian’s threshold for prescribing antimicrobials. Studies of human antimicrobial prescribing suggest similar links between prescribing and time pressure [45] and difficulty accessing further investigations [50].

Similar to the findings of a UK study [17], the Australian veterinarians interviewed felt that many of their clients had an expectation of receiving medication for a sick animal, giving them a sense that the veterinarian had ‘done something’ to fix the illness (a14, a29, b10, b13, n10) in exchange for the consultation fee. Antimicrobials can satisfy that client expectation, but in illnesses for which there are non-antimicrobial therapies available, the veterinarian can avoid antimicrobials and still give the owner ‘something … to go home with’ (c3). An example provided by more than one veterinarian was acute diarrhoea in dogs, a common presentation that has often been treated with the antimicrobial metronidazole. In recent years, a probiotic paste has been available on the Australian market that had trial evidence of efficacy in reducing the duration of clinical signs [51]. Some veterinarians said that since this product had become available, they had rarely used antimicrobials for acute diarrhoea (c3, c4, n15). Similarly, Hopman et al. reported that Dutch veterinarians cited the availability of alternative therapeutic options in deciding whether to use antimicrobials [19].

When reflecting on situations where they felt they had little choice but to give antimicrobials, interviewees often talked about unclear diagnoses, the animal having been unwell for a prolonged period (c5), especially if severely unwell (c6, b10), and situations where further diagnostics were not feasible (c6, b8), usually due to clients’ financial constraints (c7, c8, c9, c10, n1). Veterinarians felt compelled to take action to help the animal and the owner (c9) and antimicrobials often felt like the only practical option. Furthermore, once a veterinarian had mentioned the option of an antimicrobial treatment trial, it could be difficult to convince the client to do anything else (c7, n4).

Clients perceived to have lower capacity to perform home care—such as regularly cleaning a wound, monitoring relevant changes in their animal, and returning to the clinic in a timely fashion if the condition deteriorated—also sometimes made it seem impossible to withhold antimicrobials (c11, c12), particularly when there was potential for the animal to suffer (c11). However, there were some clients that interviewees felt could be trusted with such tasks, including those with a healthcare background or significant animal husbandry skills. One veterinarian said that as she had gained clinical experience, she felt more able to accurately identify those clients, and hence better able to identify opportunities to withhold antimicrobials more safely (c13).

### 3.3. Suggestions for Curbing Unnecessary Antimicrobial Use in Companion Animals

Possible solutions discussed by interviewees fell into six categories: client education; veterinarian education and training; para-veterinary staff education and training; workplace culture; informational resources; audit and sanctions.

Client education about the benefits of avoiding unnecessary antimicrobials—such as through posters in the clinic waiting room—was suggested by multiple participants, to reduce conflict with clients when withholding antimicrobials (a24, a25, a26). However, others felt that this would have limited or no impact, either because of the amount of other information clients were presented with at the clinic, or an unwillingness of clients to change their beliefs. Nonetheless, this suggestion would be relatively simple to implement and seems worthy of investigation.

Some participants suggested that increased teaching of AMS to veterinary undergraduates would be helpful; others felt that current teaching was already sufficient, but that the real-life challenges of clinical practice, including prescribing behaviour modelled by more experienced veterinarians around the new graduate, sometimes prevented veterinarians from withholding antimicrobials appropriately. Postgraduate education in antimicrobial stewardship was also suggested by some participants, with the proviso that it be delivered in a way that is convenient for busy clinicians and provided free of charge.

Strong communication skills were a clear enabler for withholding unnecessary antimicrobials while keeping clients happy. While some participants had developed these skills organically over years of trial and error, several felt that communication skills training, especially in their first few years after graduation, would have improved their antimicrobial prescribing. This suggestion would be relatively straightforward to implement and seems worthy of further research. Targeted communication skills training has been successfully trialled in general medical practitioners in Germany, and has resulted in significant reduction in inappropriate antimicrobial prescribing for upper respiratory tract infections [52].

One veterinarian suggested that client communication training for reception and nursing staff could reduce the number of difficult conversations about antimicrobials. Additionally, providing education for para-veterinary staff, explaining why unnecessary antimicrobial therapy should be avoided, as one participant had done (n15), engaged and motivated them to facilitate judicious antimicrobial use. Two modifiable aspects of practice culture were identified as enablers of appropriate antimicrobial use: inviting staff to suggest evidence-based changes to practice protocols and supporting veterinarians when they have made an appropriate decision to withhold antimicrobials.

A few participants expressed a desire for succinct, evidence-based information on common clinical conditions to provide guidance about when to prescribe antimicrobials and when to withhold them. That these veterinarians were unaware of the Australian Veterinary Prescribing Guidelines for dogs and cats, which provide this information, highlights a need to further publicise their existence. The authors have already commenced work on this task.

A few participants felt that auditing the antimicrobial use of individual veterinarians or veterinary practices, and providing feedback on their use, would be helpful. Social normative prescribing feedback has been successfully applied in other antimicrobial stewardship initiatives [53,54] and would be readily implementable in practices that submit data to a central repository, such as VetCompass (UK) and VetCompass Australia. Publicising data on actual veterinary prescribing habits could also reduce the perceptions of Australian companion animal veterinarians that their profession is using antimicrobials liberally and that a single veterinarian’s AMS efforts are therefore futile. In fact, rates of antimicrobial use in Australian pets have been declining for some years [49] and Australian veterinarians have a high degree of concern about AMR [31].

One participant, who had previously practised under an auditing system with prescriber sanctions in Denmark, felt that this was effective in curbing unnecessary use. However, they also acknowledged the difficulty of executing such an intervention in Australia, as this would require legislative changes and a mechanism to monitor antimicrobial use in animals.

### 3.4. Further Discussion

To our knowledge, this study is the first to reveal the pivotal role of client capacity in the choice to withhold antimicrobials. Clients who were perceived as motivated and capable (of undertaking non-antimicrobial management and monitoring) enabled veterinarians to appropriately withhold antimicrobial treatment. This finding suggests the potential for AMS interventions targeting pet owner skills and motivation.

This study also documents for the first time the roles of clients’ gender and age prejudices, veterinary workplace culture and para-veterinary staff in appropriate use of antimicrobials. Veterinarians who felt supported by their colleagues and who felt their workplaces were open to change were empowered to withhold antimicrobial treatment. Para-veterinary staff could either undermine or support AMS efforts through their conversations with clients and with the veterinarians themselves. This underlines the importance of involving receptionists, veterinary nurses and practice managers in antimicrobial stewardship initiatives. Cultivating a supportive and respectful workplace culture, where young and female veterinarians are publicly valued, could help to tackle clients’ gender and age prejudices and reduce unnecessary prescribing.

The influence of social media on veterinary decision-making, or, more precisely, the threat of negative comments on the practice’s social media page, is another new finding. This is also the first study that we are aware of to link the antimicrobial stewardship messages of human physicians with the behaviour of their patients as veterinary clients, suggesting that a united public education campaign about antimicrobials could yield benefits to all sectors.

A major strength of this study was the candour of the interviewees. This was likely aided by the interviewer’s role as a fellow veterinarian, which established trust and empathy with participants, and her ability to suggest realistic variations on scenarios to draw out veterinarians’ beliefs. Empathic interpretation of the interview transcripts was also enhanced by the authors’ lived experiences. Additionally, although the authors had expected some findings to be specific to the local context, the themes either echoed findings of previous UK and Dutch studies or were novel themes that are likely to exist in companion animal medicine in many other countries. Another strength is the application of a widely used theoretical framework to the thematic analysis, allowing for simpler comparison with other qualitative study findings. A limitation is that only three of the 22 participants expressed a low interest in AMR and AMS, while ten stated that they had a high interest in AMR and AMS. Thus, solutions suggested by participants to reduce inappropriate prescribing may be more suited to veterinarians who are already somewhat engaged with AMS, and less suitable for those with low interest.

## 4. Conclusions

Many veterinarians are motivated to use antimicrobials only when they are necessary, as they are worried about the consequences of AMR. However, these concerns can be outweighed by behavioural beliefs, especially fears of clinical deterioration and client dissatisfaction if antimicrobials are withheld or delayed. These fears can be compounded by diagnostic uncertainty and time pressure, as well as a range of client factors, including client expectations of receiving medication, financial constraints, a poor veterinarian–client relationship and doubts about the client’s capacity to nurse and monitor their animal. Prescribing antimicrobials often gives veterinarians a sense of safety; conversely, withholding or delaying antimicrobial treatment can feel like an act of bravery.

Access to evidence-based prescribing guidelines, a supportive and change-ready workplace culture, para-veterinary staff who are engaged in judicious antimicrobial use, and strong communication skills were all found to encourage appropriate withholding of antimicrobial treatment. These enablers can be incorporated into veterinary undergraduate and continuing education. Uptake of pet insurance and availability of alternative, non-antimicrobial therapeutic products also enabled antimicrobial withholding.

Veterinary antimicrobial stewardship initiatives could benefit from consideration of these complex influences on a veterinarian’s decision to withhold or delay antimicrobial use, and from the suggestions made by veterinarians in this study to reduce unnecessary antimicrobial use, such as social normative feedback on their prescribing, client AMS education and veterinarian communication skills training.

## Figures and Tables

**Figure 1 antibiotics-12-00540-f001:**
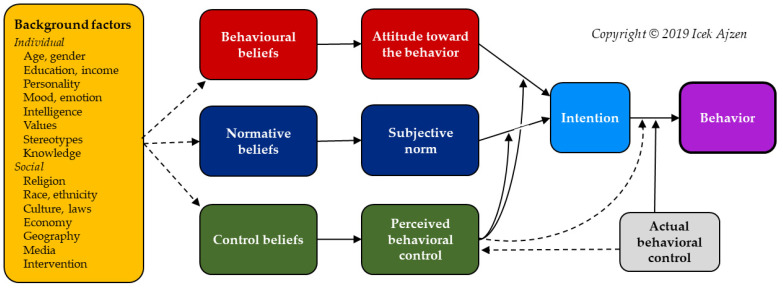
Icek Ajzen’s Theory of Planned Behaviour (TPB) with background factors was used to organise the thematic analysis. Reproduction of this figure is permitted in journal articles, as stated on the University of Massachusetts website from which it is sourced: https://people.umass.edu/aizen/tpb.background.html (accessed on 12 July 2022).

**Figure 2 antibiotics-12-00540-f002:**
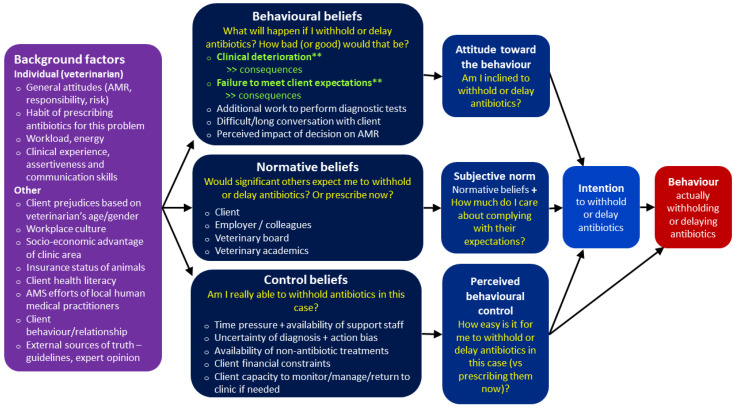
Summary of factors influencing veterinarians’ decisions to withhold or delay antimicrobial treatment in the absence of a clear need, using an adaptation of the Theory of Planned Behaviour shown in Figure 1. ** Behavioural beliefs about clinical deterioration and failure to meet owner expectations were key themes and are further explored in Section 3.2.2.

**Figure 3 antibiotics-12-00540-f003:**
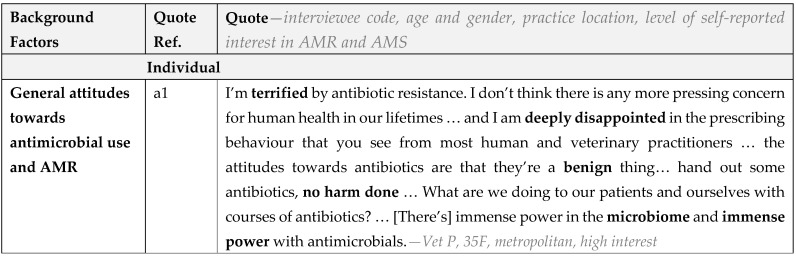
Background factors influencing the decision to withhold or delay antimicrobial treatment where there was no clear indication, and illustrative quotes. Bold text has been used to highlight key ideas. Grey italics have been used for participant code, age and gender, location of practice and level of interest in AMR and AMS. *** represents redacted obscenity.

**Figure 4 antibiotics-12-00540-f004:**
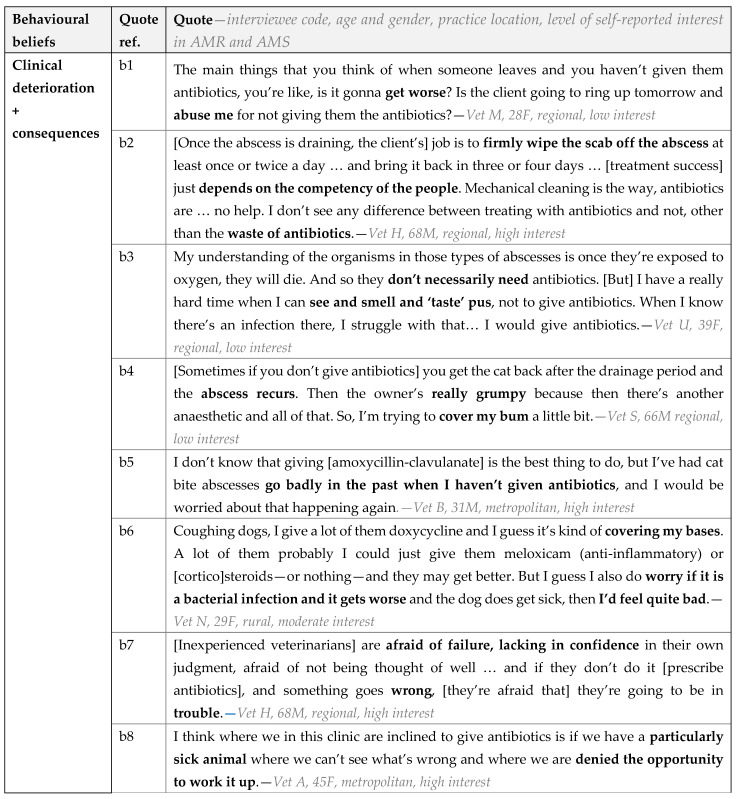
Background factors influencing the decision to withhold or delay antimicrobial treatment where there was no clear indication, and illustrative quotes. Bold text has been used to highlight key ideas. Grey italics have been used for participant code, age and gender, location of practice and level of interest in AMR and AMS. There were three main scenarios in which participants described giving systemic antimicrobials when they were not indicated. The first is where the cause of illness was probably or certainly not responsive to antimicrobials, but the client expected the veterinarian to medicate the animal, sometimes specifically with an antimicrobial (b1, b7, b11, b12, b13, b15, a29). Alternatively, it could be where the client was frustrated (a27) or despondent about their animal’s condition, such as the client ‘at their wit’s end’ dealing with a dog with diarrhoea (a28), increasing the pressure for the veterinarian to take action to address the problem.

**Figure 5 antibiotics-12-00540-f005:**
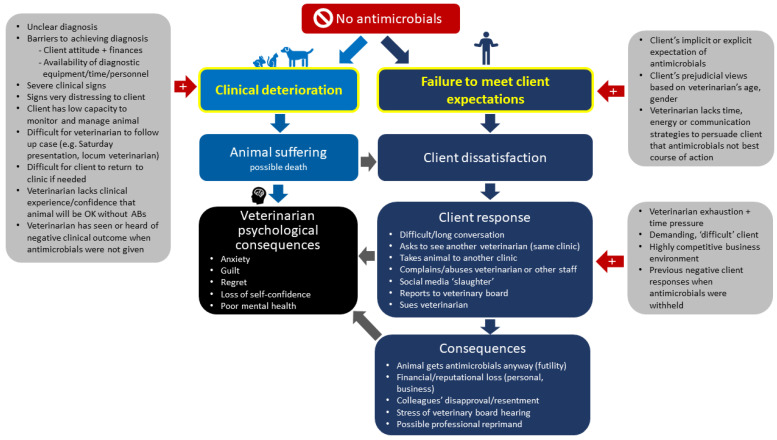
Schema of the dominant behavioural beliefs that interviewees associated with withholding or delaying antimicrobial treatment. Secondary factors that heighten fears are shown in grey.

**Figure 6 antibiotics-12-00540-f006:**
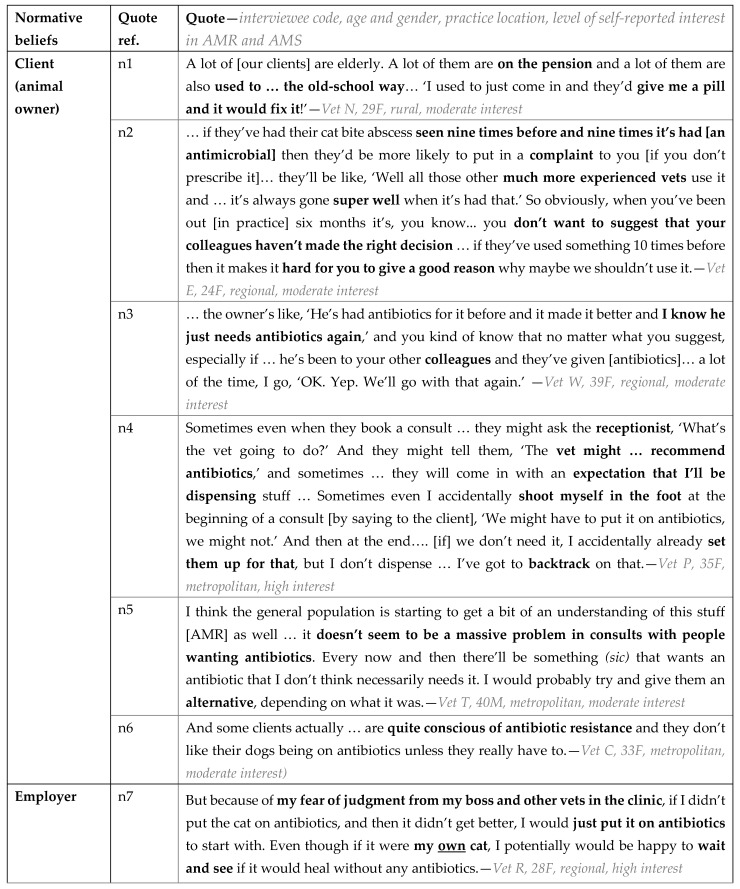
Normative beliefs influencing decision to withhold or delay antimicrobial treatment where there is no clear indication, and illustrative quotes. Bold text has been used to highlight key ideas. Grey italics have been used for participant code, age and gender, location of practice and level of interest in AMR and AMS. As previously discussed, multiple interviewees mentioned that their clients commonly communicated an expectation of receiving antimicrobials for their animal, which made it more difficult to withhold them. Such expectations were often established by other veterinarians who had previously prescribed antimicrobials (n1, n12), especially for the same clinical presentation (n2, n3). In this situation, withholding antimicrobials could be seen as directly contradicting the previous veterinarian(s), which led to psychological discomfort, particularly when the previous veterinarian was more experienced (n2). Veterinarians in the UK also cited this awkwardness as a driver of antimicrobial prescribing [18]. In human hospitals, a similar unwillingness to break professional etiquette by questioning the decisions of others, particularly others of higher status, also enables inappropriate antimicrobial prescribing [46,47].

**Figure 7 antibiotics-12-00540-f007:**
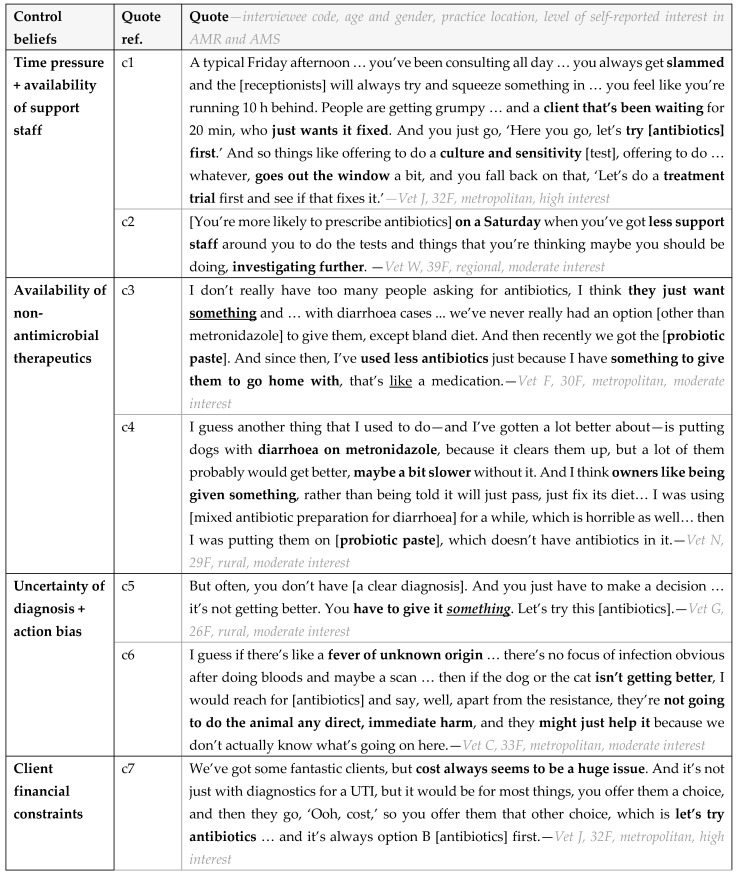
Control beliefs influencing decision to withhold or delay antimicrobial treatment where there is no clear indication, and illustrative quotes. Bold text has been used to highlight key ideas. Grey italics have been used for participant code, age and gender, location of practice and level of interest in AMR and AMS.

**Table 1 antibiotics-12-00540-t001:** Participant characteristics (n = 22).

Characteristic	Category	n	%
Gender	Female	16	73%
	Male	6	27%
Age	20–29	6	27%
	30–39	10	45%
	40–49	3	14%
	50–59	1	5%
	60–69	2	9%
Postgraduate veterinary qualifications	Yes	5	23%
	No	17	77%
Role	Principal Veterinarian	9	41%
	Associate/Senior Associate	10	45%
	Locum or other	2	9%
Practice location	Metropolitan	10	45%
	Regional	9	41%
	Rural	3	14%
Practice type	Primary/emergency care only	19	86%
	Specialist/referral	1	5%
	Combination of above	2	9%
Animals serviced	Small animals only	19	86%
	Mixed practice	3	14%
Size of practice (veterinary full-time equivalent staff)	2 to 3 veterinary FTEs	6	27%
	4 to 5 veterinary FTEs	11	50%
	more than 5 veterinary FTEs	5	23%
State where practice is located	Victoria	10	45%
	New South Wales	6	27%
	Queensland	2	9%
	South Australia	2	9%
	Tasmania	1	5%
	Western Australia	1	5%
Self-reported interest in AMR and AMS	Low	3	14%
	Moderate	9	41%
	High	10	45%

## Data Availability

The raw data are not publicly available due to restrictions of this project’s ethical approval.

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
