# Peer review of "‘Brave Enough’: A Qualitative Study of Veterinary Decisions to Withhold or Delay Antimicrobial Treatment in Pets"

_antibiotics, 2023, doi:10.3390/antibiotics12030540_

Round 1

Reviewer 1 Report

Major comments

- Methods section should be reconstructed in subsection e.g. participants, survey, etc. 

- Basic information of participants should be visualized instead of using texts as authors provided in lines 129-140.

- Authors used several words like "most participants", "some", "Almost all interviewees. How many percentage or how many of them, please provide the supplementary data or some table for this. 

- Significant of this study and future implementation should be more discussed and provided in the discussion and conclusion. 

Minor comments

1.     Line 37: “This process is accelerated under the selection pressure of anti-bacterial antimicrobial agents [2], which are commonly referred to as antibiotics.” could be replaced by “This process is accelerated under the selection pressure of antimicrobial agents [2].”.

2.     Line 86: The full name of “AMS” should be provided as first mentioned.

3.     Lines 113, 115, 119, and 641: “RS” – “R.S.”

4.     Line 115: “AS” – “A.S.”

5.     Line 139: “Error! Reference source not found.” – please check this.

6.     Lines 536 and 588: “2.5” and “2.6” should be “3.5” and “3.6”, respectively.

7.     Lines 537 – 539: The authors mentioned that there were 7 categories of possible solutions discussed by interviewees, but the authors provided only 6 categories. Please check this issue.

8.     Figure 1 should be provided in better quality (resolution).

Reviewer 2 Report

The topic of this manuscript is interesting, concerns the views of veterinarians in regards to an emerging problem with antimicrobial therapy.

The authors are up to a wide variety of factors and beliefs to organize especially when this is a multifactorial topic.

I recommend some changes to be made regarding the general aspect of the manuscript.

Line 37-38 should be re-written in better English

Lines 43-46 should be re-written presenting the problem without leaving an issue for the prescribers that are purposely misdiagnose (line 44) ….and the antimicrobials cannot change their target group by narrowing the spectrum and most have more than one bacteria as a target (line 45)

As authors suggest in their introduction paragraph (lines 47-62) this is mostly an Australian research and much are different comparing to England and to EU and USA professionals…so I believe that this should be included in the article title even though some may be common…i.e an Australian study

Also another limitation of this study is the lack of large animal practice veterinarians, which their views are more important as the use of antimicrobials is vastly larger in amount and affecting food industry and public health due to consumption.

Moreover, another limitation to the study is the small number of 22 interviews comparing to the number of total vets of small animal practices in Australia, even if 1 vet per practice is used. Nonetheless, the general attitude is that of interest.

Also, line 86, AMS initials should be explained
